# Speckle and Shadows: Ultrasound-specific Physics-based Data Augmentation for Kidney Segmentation

**Rohit Singla**                                                                RSINGLA@ECE.UBC.CA
**Cailin Ringstrom**                                                       CERINGSTROM@ECE.UBC.CA
**Ricky Hu**                                                                        RHU@ECE.UBC.CA
**Victoria Lessoway**
**Janice Reid**
**Robert Rohling**                                                          ROHLING@ECE.UBC.CA
[1] *Electrical and Computer Engineering, 2332 Main Mall, Vancouver, Canada, V6T 1Z4*
**Christopher Nguan**                                       CHRIS.NGUAN@UBCUROLOGY.COM
[2] *Urologic Sciences, 2775 Laurel St, Vancouver, Canada, V5Z 1M9*

**Editors:** Under Review for MIDL 2022

## Abstract

Techniques for data augmentation are widely employed to avoid overfitting, improve generalizability and overcome data scarcity. This data-oriented approach frequently uses domain-agnostic approaches such as geometric transformations, colour space transformations, and generative adversarial networks. However, utilsing domain-specific characteristics in augmentations may result in additional invariances or improved robustness. We present several augmentation techniques for ultrasound: zoom, time-gain compensation, artificial shadowing, and speckle parameter maps. Zoom and time-gain compensation mimic traditional image quality parameters. For shadowing, we characterize acoustic shadows within abdominal ultrasound images and provide a method for incorporating artificial shadows into existing images. Finally, we transform B-mode ultrasound images into Nakagami-based speckle parameter maps to describe spatial structures that are not visible in conventional B-mode. The augmentations are evaluated by training a fully supervised network and a contrastive learning network for multi-class intra-organ semantic segmentation. Our preliminary results reflect the difficulties of creating augmentations as well as the limitations posed by acoustic shadowing.

**Keywords:** Ultrasound, temporal coherency, speckle, contrastive learning, segmentation

## 1. Introduction

While tremendous progress has been made as a result of unique architectures, efficient large-scale computational power, and the availability of data, machine learning techniques continue to demand growing volumes of data. The more data that is provided, the more successful the model, and the less likely it will overfit. Regularization, dropout, batch normalisation, and transfer learning are all methods for preventing over-fitting in situations with insufficient data, such as medical imaging. Another solution is data augmentation.

This data-oriented approach frequently incorporates domain-agnostic techniques for creating new images and/or labelling from existing data. This includes geometric changes such as translation, rotation, flipping, scaling, warping or deformation, and intensity transformations such as occlusion and Gaussian noise insertion. Additionally, there are ways to use

deep learning such as generative adversarial networks or varational auto-encoders. (Pang et al., 2021; Pesteie et al., 2019; Zaman et al., 2020) The application of these augmentations has demonstrated potential improvements in task accuracy.(Perez and Wang, 2017)

Medical imaging modalities such as magnetic resonance imaging, computed tomography, and ultrasound, all exhibit characteristics unique from natural images. For instance, an ultrasound image is relies on time-of-flight for radio-frequency waves rather than light. When acoustic waves enter the body, they interact with the subsurface tissue structures. Different interactions will result in a change in the processed ultrasound image. Taking these distinctions into account, exploiting unique domain characteristics may result in augmentations that reflect previously overlooked or difficult data. For example, k-space sampling is used in augmenting magnetic resonance images (Liu et al., 2019) and acquisition parameters are modified in computed tomography(Omigbodun et al., 2019); both methods enable the creation of challenging data for a variety of tasks.

Recently, researchers have begun to examine how ultrasound-specific features might be employed to develop techniques for creating new augmentations.(Lee et al., 2021) and (Tirindelli et al., 2021) are recent examples. (Lee et al., 2021) develop a systematic technique for learning the ideal set of transformations to improve view classification. Their collection comprises 18 unique transformations such as grid distortion, elastic transform, and speckle. Their method trumps human selection, demanding the production of a bigger collection of possible changes. The authors provide deformation, reverberation, and signal-to-noise ratio as novel methodologies for bone segmentation and classification in (Tirindelli et al., 2021). While the initial results were modest improvements and necessitated prior knowledge of the region of interest, their approach is one of the first to depart from established strategies and incorporate understanding of the physics underlying ultrasound generation.

We propose three new contributions to ultrasound data augmentation based on these efforts and their physics-based and principled approaches: (1) zoom, (2) acoustic shadowing created artificially, and (3) speckle parameter maps. Zoom magnifies the ultrasound image while keeping the form of the ultrasound sector. Artificial acoustic shadowing augments existing ultrasound images with one of the most often reported imaging artefacts in ultrasound using a statistical approach. The speckle parameter maps are inspired by the concept of colour spaces and show previously unseen spatial patterns in conventional B-mode ultrasonography. The three augmentations are tested in training two architectures for multi-class intra-organ semantic segmentation.

## 2. Methods

While several parameters of an ultrasound machine can be configured, our selection of augmentations is based on (i) being organ-agnostic, allowing broad use across all ultrasound domains and (ii) rethinking how imaging artefacts how machine learning models.

**Zoom and Time-Gain Compensation.** Zoom and time-gain compensation are ones that sonographers often adjust manually to optimize the image quality. By modelling these, simulated images reflect a range of the different image qualities. The zoom augmentation is a third-degree spline interpolation on the image's centre, upscaling by a factor between 0 and 2.5. The upscaled image is masked to keep the sector shape. The zoomed image loses spatial resolution compared to the original. In ultrasound, time-gain compensation (TGC)

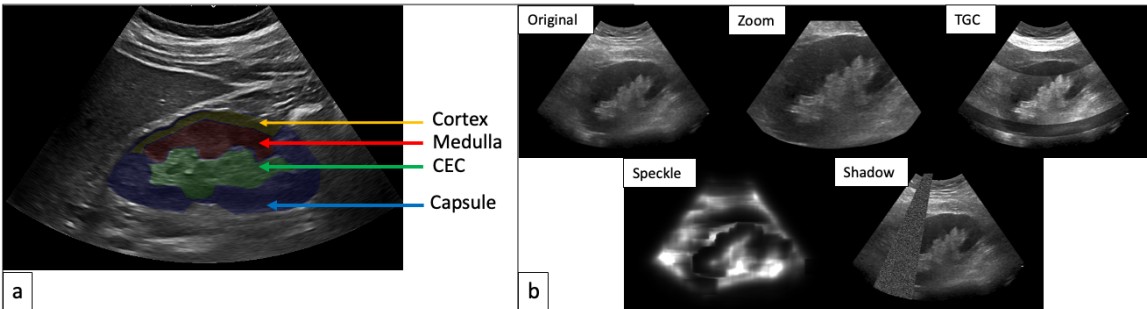

Figure 1: An example image with the four augmentations presented. Note the speckle augmentation here is the Nakagami shape parameter map.

is a way to reduce the impact of attenuation. To simulate it, we divide the sector into 10 crescent shaped segments. Ten is a common number of segments found in ultrasound; other numbers are possible. Each segment is randomly saturated between [0.5,2].

**Acoustic Shadow.** Acoustic shadowing is one of the most often encountered artefacts in ultrasound. Shadows in an image may be caused by insufficient contact between the transducer and the surface or locations with large acoustic impedance variations between tissue interfaces. While acoustic shadows are similar to occlusions, they are not zero. The pixels in shadow regions follow a Nakagami probability distribution.(Hu et al., 2019a)

The Nakagami distribution is a left-skewed distribution characterized by two parameters: shape $m$ and scale $\Omega$. Given an image $x$, it's Nakagami distribution is defined as in (Equation (1)). $\Gamma(*)$ indicates the Gamma function.

$$N(x; m, \Omega) = \frac{2m^m}{\Gamma(m)\Omega^m} x^{2m-1} e^{\frac{-m}{\Omega} x^2} \tag{1}$$

(Hu et al., 2019b) were able to increase ultrasound segmentation performance by utilising this shadow detection as an attention mechanism in deep learning. We hypothesize that augmenting with shadows may improve the network by making it invariant to such artefacts.

We randomly sample 61 abdominal ultrasound images with an acoustic shadow artifact present. This data is separate and not included in any model training or evaluation. Acoustic shadows are manually segmented. A Nakagami distribution was fitted to each region in each image $I$ using the estimators from (Kolar et al., 2004), as in Equation (2) and Equation (3). These estimators make use of the image's expectation $E$ and variance $Var$. After averaging the resulting values, a reference Nakagami distribution for abdominal acoustic shadows is created. The mean shape was $m = 0.202$ and mean scale was $\Omega = 189.3$.

To generate the shadows, two parameters are required: the scanline indicating the shadow's centre $i$ and the width $w$. The boundaries of the ultrasound are determined via edge detection and then simulated as two intersecting lines. Identifying this point of intersection enables the conversion of Cartesian to polar coordinates, which is required for curved transducers, the most common type of transducer for abdominal imaging. Within these boundaries, from the point of intersection, an angular position and width are randomly

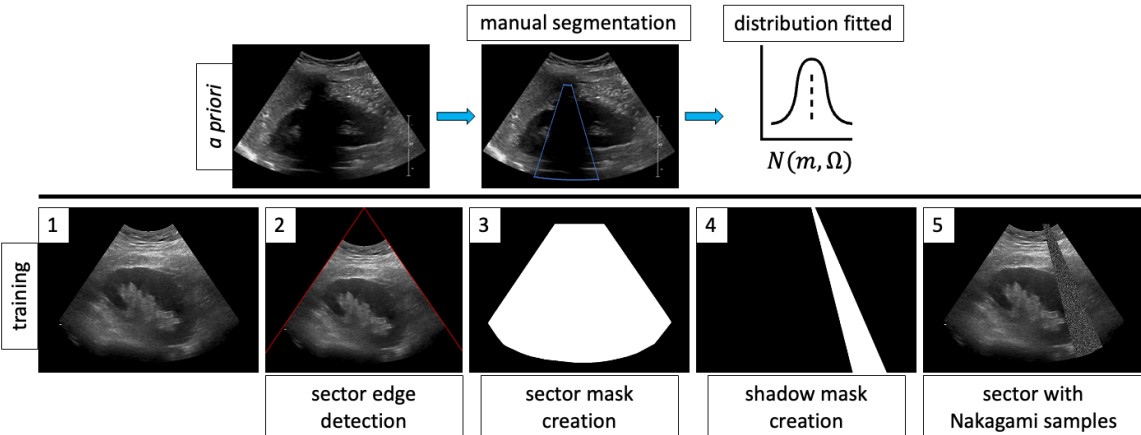

Figure 2: Constructing an acoustic shadow. Prior to any experiments, a set of shadows are manually segmented in a separate dataset. The mean Nakagami distribution is found. During training, the ultrasound sector mask is calculated through edge detection (2 and 3). An randomly selected scanline and width great a new mask (4). The mask's values are randomly drawn from the mean Nakagami distribution previously found, and applied to the original image (5).

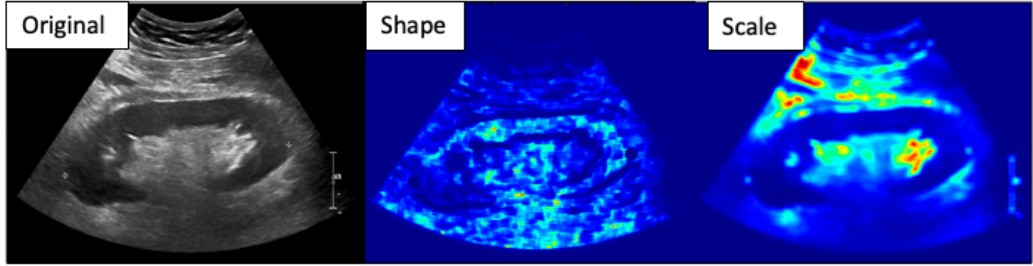

Figure 3: Example images of (left) a B-mode image, (middle) the Nakagami shape parameter map, and (right) the scale parameter map. Note how the shape map highlights features in the kidney not readily visible in B-mode.

selected. This defines a sector-sized mask. This mask's pixel values are replaced with values drawn from $(N(x; 0.202, 189.3))$ and randomly placed in the image. Values beyond the [0,255] range were excluded and re-sampled. The ultrasound sector's boundaries are used to retain the original shape.

**Speckle Parameter Maps.** A parameter maps are generated by sliding a window across the image and calculating some metric on it. The center of the patch is assigned the result. In this manner, these maps are transformations of the original image. In ultrasound, using the Nakagami parameters to generate parameter maps may aid in segmentation. This

is due to the the inherent relationship between the Nakagami distribution and the underlying tissue structures. We transform the ultrasound into "speckle space". First, we utilise a patch size of 20x20. The Nakagami parameters are fitted in each patch. Two output images are generated, one for each of the $m$ and $\Omega$ Nakagami parameters. The generated images are illustrated in Figure 3. While the ideal patch size calculation uses axial and lateral resolutions to compute the smallest window with relevant speckle (correlation length), we approximate it. The correlation length is on the order of a resolution cell.(Wagner, 1983) (Byra et al., 2016) states that a window three times the pulse length is sufficient for speckle analysis. We can estimate the correlation length because the pulse length is on a similar order of magnitude to the correlation length. Given that the resolution cell of a curvilinear ultrasound transducer is typically 0.1 to 0.2 mm in diameter, and the image has an estimated pixel to millimetre ratio of 1 to 0.01, we choose a patch size of 20x20.

We view these maps as pseudo-labels, and use them to create new positive pairs in a contrastive network for medical image segmentation.(Chaitanya et al., 2020) Rather than selecting both parameter maps for augmentation, we follow the lead of (Tian et al., 2020) and choose the parameter map that is most likely to increase performance by supplementing the trained model with additional information. We compute the average mutual information of the B-mode image with each of the parameter maps and select the map with the minimum mutual information.

$$m = \frac{E[I^2]^2}{Var[I^2]} \qquad (2) \qquad\qquad \Omega = E[I^2] \qquad (3)$$

### 2.1. Data Set and Experiments

Following permission by Institutional Research Ethics Board (H19-02669), 514 kidney ultrasound pictures were retrieved from the same number of patients(Vancouver General Hospital, Vancouver, BC). Manually prepared fine-grained polygon annotations for four classes (kidney capsule, cortex, medulla, and central echogenic complex) by two sonographers with a combined experience of more than 40 years. Approximately 7000 videos with approximately 200 frames were used for the pre-training in contrastive learning approaches. Curvilinear transducers were used. Imaging depths ranged from 5 to 16 cm.

We conduct three experiments. We employ nnU-net from (Isensee et al., 2021) as a representation of the state-of-the-art in fully supervised medical image segmentation in the first experiment. This network is data-adaptive, analysing the data set first (prior to augmentations) and then adjusting its training approach accordingly. We apply the zoom, TGC, shadow augmentations on the nnU-net, utilising all labelled data. Default pre-processing, training strategies and hyperparameters are used. A two-dimensional U-Net is trained for 500 epochs with an Adam optimizer, a combined loss of DSC and cross-entropy, a learning rate starting at 0.01 and annealed during training. Our augmentations had 20% chance of being applied to an image. In the second, we apply the contrastive learning segmentation network from (Chaitanya et al., 2020) to ultrasound images. Unlabelled image-level pairs are sampled as similar (positive) or dissimilar (negative) pairs to pre-train an encoder in the first step of this approach. The second stage pre-trains a decoder by sampling patches within images rather than complete images. These two steps use NT-Xent loss to consider augmentations as pseudo-labels. The batch size is 64. The third step is trained in a supervised manner, a DSC loss, and a batch size of 12. All steps use an Adam optimizer with a

Table 1: The results of incorporating augmentations into the nnU-net from (Isensee et al., 2021). Measures are reported as average (DSC, HD), with the best value in bold. HD is reported in millimetres. For each column, the highest value is in bold and * indicates statistical significance of a two-tailed t-test (p ≤ 0.05).

| Augmentation | Capsule | CEC | Medulla | Cortex | All |
|---|---|---|---|---|---|
| nnU-net | 0.85, **9.7** | 0.74, **8.4** | 0.56, 10.2 | 0.52, 13.0 | 0.67, **10.4** |
| + Zoom | **0.86**, 10.8 | **0.77**, 8.6 | 0.55, 10.5 | 0.51, 15.4 | 0.67, 11.3 |
| + TGC | 0.83, 10.6 | 0.76, 9.3 | **0.58**, 10.7 | **0.54, 12.8** | **0.69**, 10.9 |
| + Shadow | 0.82, 11.6 | 0.75, 9.7 | 0.49, 11.6 | 0.46, 13.4 | 0.63, 11.7 |

learning rate of 0.001, and 10,000, 5000, and 10,000 epochs, respectively. We train models with standard augmentations then with each one of our proposed algorithms. This is performed for 1%, 10%, 50%, and 100% of labels. Finally, we investigate changes in network uncertainty. Our uncertainty metric is the averaged variance after Monte Carlo dropout for 500 iterations.(Camarasa et al., 2020) The results are expressed as the mean Dice-Sørensen coefficient (DSC), Hausdorff distance (HD) or averaged variance (AV). Augmentations code is available at https://github.com/rsingla92/speckle_n_shadow.

## 3. Results

The average mutual information for the shape parameter map was 0.40, and was 0.60 for the scale parameter map. All subsequent tests utilise the Nakagami shape parameter map as an augmentation. Table 1 summarises Experiment 1 (augmentations in fully supervised learning). In the fully supervised setting, the state-of-the-art nnU-net did not statistically significantly differ in its DSC or HD values across all four classes as compared to default augmentations. Table 2 summarises Experiment 2 (augmentations in contrastive learning). Figure 4 compares the segmentations graphically. Table 3 summarises the uncertainty experiments. TGC has significantly improved uncertainty on most classes for the nnU-net while it has significant worse uncertainty in the contrastive network.

## 4. Discussion and Conclusion

We study zoom, TGC, artificial acoustic shadow and speckle parameter maps in this work, demonstrating the effects of such augmentations on DSC, HD, and AV in multi-class intra-organ ultrasound semantic segmentation. Unlike other works, our augmentations require no knowledge of the underlying tissues or transducer. The preliminary results are inconclusive. Individually, the added augmentations demonstrated mixed results.

First, we observe no statistically significant changes in DSC or HD when adding our augmentations to nnU-net. Only augmenting with TGC, an intensity based augmentation, significantly improved uncertainty in three of four classes. In the contrastive network, we observe no clear pattern. In certain models and classes, an improvement is observed. However, no augmentation provides consistent benefit to DSC or HD. The impact of TGC on uncertainty in contrastive learning was the opposite than in nnU-net as three classes

had worsened uncertainty. The differences may be in how supervised methods learn in label space, whereas contrastive ones learn representation. The mixed results highlight that designing effective augmentations which add meaningful invariances without sacrificing accuracy is difficult to perfect; an augmentation may benefit one paradigm but not the other.

Second, shadow augmentation was consistently inferior. Because we ignored the shadow's spatial structure, our shadows spanned the entire image vertically. Small shadows at tissue interfaces in the image may help, such as in bones. Acoustic shadowing is based on an *a priori* known Nakagami distribution of shadow artefacts. Its parameter estimation may be improved by optimising the patch size or using the raw radiofrequency data rather.

Third, the nnU-net has a mean DSC of 0.67. Given this relatively low value, and knowing the cortex and medulla classes constitute 5-10% of the total image, we explored how sensitive DSC would be to small changes. A 10-pixel erosion/dilation on the ground truth caused up to a 30% change in DSC. Fourth, augmentations can be unrealistic augmentations,such as cutmix or mixup which demonstrated performance gains.(French et al., 2019; Chen et al., 2019) Consideration of how augmentations like shadow and speckle could be incorporated in unrealistic manners or in local symmetries may be worthwhile for ultrasound augmentations.(Winkels and Cohen, 2019)

Additional ultrasound data sets will be used in future research to compare physics-based and learned augmentations. Alternatively, including our augmentations in the search strategy from (Lee et al., 2021) may prove fruitful. In conclusion, we contribute a) a time-gain compensation augmentation which shows decreased uncertainty in supervised settings and b) an acoustic shadow distribution and method to create artificial shadows applicable for

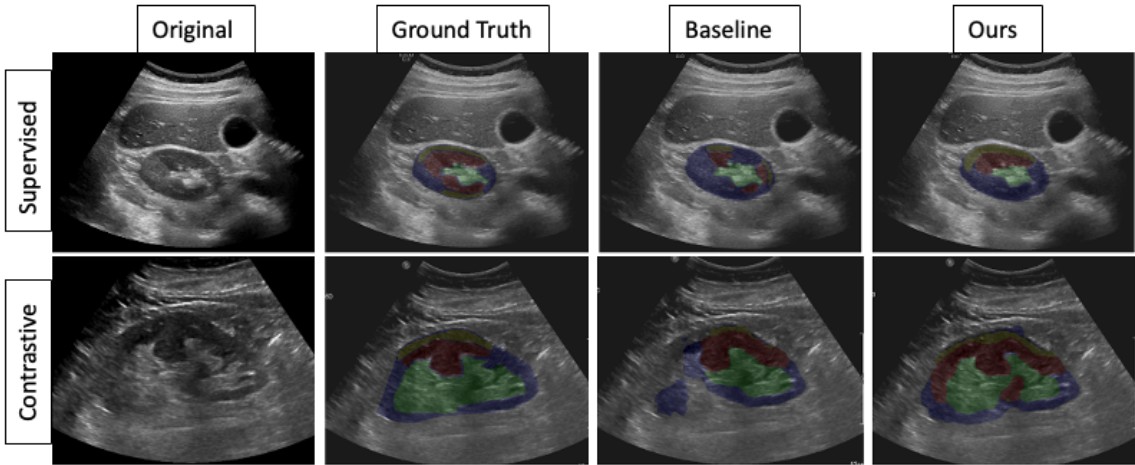

Figure 4: Predicted segmentation mask visualisation. From left to right: the original B-mode image, the expert's ground truth, the network's result without our augmentations, and the network's result with them. The first row shows a nnU-net with zoom and shadow augmentations, and the second a contrastive network with 50% of labels and the speckle parameter map augmentation.

Table 2: The results of augmentations in the contrastive network using different amounts of data. Measures are (mean DSC, mean HD). HD is in millimeters. For each block, the highest value is in bold and statistical significance indicated by *.

| Augmentation | Percent | Capsule | CEC | Medulla | Cortex | All |
|---|---|---|---|---|---|---|
| Chaitanya *et al.* | 1% | 0.52, **27.3** | 0.29, 32.8 | 0.21, 25.0 | 0.13, 27.0 | **0.29**, 28.0 |
| + Zoom | 1% | 0.38, 53.6*, | 0.29, 39.4* | 0.24, 27.7 | 0.13, **26.5** | 0.26, **26.8**\* |
| + TGC | 1% | **0.56**\*, 30.6 | **0.34**\*, 35.9 | 0.19, 32.0* | 0.05*, 33.2 | 0.28, 33.0 |
| + Shadow | 1% | 0.51, 31.0 | 0.17*, 30.0 | 0.24, **22.6** | **0.15**, 29.8 | 0.27, 28.4 |
| + Speckle Map | 1% | 0.42*, 31.6 | 0.27, **26.1** | **0.25**\*, 22.9 | 0.10, 26.6 | 0.26, **26.8**\* |
| Chaitanya *et al.* | 10% | **0.75**, 18.9 | **0.59**, **15.0** | 0.31, **15.4** | 0.20, 19.2 | **0.47**, **17.1** |
| + Zoom | 10% | 0.73, **16.6**\* | 0.50, 23.7* | 0.25*, 19.6* | 0.20, 19.3 | 0.42, 19.8 |
| + TGC | 10% | 0.74, 16.7* | 0.46, 17.8 | 0.32, 16.2 | **0.25**\*, 19.0 | 0.44, 17.4 |
| + Shadow | 10% | 0.74, 23.7 | 0.45*, 18.6 | 0.26, 15.7 | 0.15*, 21.1 | 0.40, 19.8 |
| + Speckle Map | 10% | 0.71, 19.6 | 0.44*, 21.2* | **0.32**, 16.5 | 0.19, **18.0** | 0.41*, 18.6 |
| Chaitanya *et al.* | 50% | 0.80, 13.7 | **0.65**, **12.5** | **0.36**, 14.2 | 0.26, 16.1 | **0.52**, **14.1** |
| + Zoom | 50% | 0.81, **12.4** | 0.64, 14.0 | 0.35, **14.0** | **0.31**\*, 15.8 | **0.52**, **14.1** |
| + TGC | 50% | **0.84**, 13.5 | 0.62, 18.8* | 0.32, 16.7 | 0.26, 18.6 | 0.50, 16.9 |
| + Shadow | 50% | 0.79, 14.6 | **0.65**, 17.2 | 0.34, 15.3 | 0.24, **14.0** | 0.50, 15.3 |
| + Speckle Map | 50% | 0.81, 13.4 | 0.61, 17.6 | 0.34, 16.5 | 0.27, 21.3 | 0.51, 17.2 |
| Chaitanya *et al.* | 100% | **0.82**, 11.3 | 0.67, 12.8 | 0.38, 15.7 | 0.30, 24.8 | 0.54, 16.1 |
| + Zoom | 100% | **0.82**, **10.9** | 0.67, 10.7 | **0.40**, 12.4 | **0.32**, 14.8 | **0.55**, 12.2 |
| + TGC | 100% | **0.82**, 11.0 | 0.67, 11.5 | 0.37, 16.6 | **0.32**, 16.4 | 0.54, 13.9 |
| + Shadow | 100% | 0.80, 14.0 | 0.65, 11.0 | 0.37, 11.9 | **0.32**, 14.2 | 0.54, 12.8 |
| + Speckle Map | 100% | 0.80, 13.7 | **0.70**, 9.0* | 0.37, **11.3**\* | 0.30, **14.0**\* | **0.55**, **12.0**\* |

Table 3: The averaged variances from adding augmentations into nnU-net and a contrastive network. Lowest values bolded. * indicates statistical significance ($p \leq 0.05$).

| Augmentation | Capsule | CEC | Medulla | Cortex | All |
|---|---|---|---|---|---|
| nnU-net | $\mathbf{3.2 \times 10^{-6}}$ | $9.7 \times 10^{-7}$ | $1.5 \times 10^{-6}$ | $1.3 \times 10^{-6}$ | $1.8 \times 10^{-6}$ |
| + Zoom | $\mathbf{3.2 \times 10^{-6}}$ | $1.4 \times 10^{-6}$ | $1.9 \times 10^{-3}$ | $2.1 \times 10^{-6}$ | $2.2 \times 10^{-6}$ * |
| + TGC | $4.9 \times 10^{-6}$ | $\mathbf{5.3 \times 10^{-7}}$* | $\mathbf{4.8 \times 10^{-7}}$* | $\mathbf{6.0 \times 10^{-7}}$* | $\mathbf{1.6 \times 10^{-6}}$ |
| + Shadow | $5.4 \times 10^{-6}$ | $6.8 \times 10^{-7}$ | $1.1 \times 10^{-6}$ | $1.2 \times 10^{-6}$ | $2.1 \times 10^{-6}$ |
| Chaitanya *et al.* | $2.6 \times 10^{-4}$ | $5.1 \times 10^{-4}$ | $\mathbf{5.4 \times 10^{-4}}$ | $1.1 \times 10^{-3}$ | $5.9 \times 10^{-4}$ |
| + Zoom | $\mathbf{2.5 \times 10^{-5}}$* | $6.0 \times 10^{-4}$* | $5.5 \times 10^{-4}$ | $9.2 \times 10^{-4}$* | $5.9 \times 10^{-4}$ |
| + TGC | $2.3 \times 10^{-4}$* | $7.4 \times 10^{-4}$* | $9.4 \times 10^{-4}$* | $\mathbf{1.3 \times 10^{-4}}$* | $8.1 \times 10^{-4}$* |
| + Shadow | $2.6 \times 10^{-4}$ | $\mathbf{4.4 \times 10^{-4}}$* | $6.2 \times 10^{-4}$* | $1.0 \times 10^{-3}$ | $5.9 \times 10^{-4}$ |
| + Speckle Map | $2.7 \times 10^{-4}$ | $4.6 \times 10^{-4}$* | $5.7 \times 10^{-4}$ | $6.7 \times 10^{-4}$* | $\mathbf{4.9 \times 10^{-4}}$* |

abdominal imaging, and c) a thorough and systematic approach to evaluating augmentations in supervised and contrastive settings. Exploring how inherent ultrasound properties, such as speckle and shadow, can be used to optimise task performance or robustness is an existing avenue for research.

## Acknowledgments

The authors thank the Natural Sciences and Engineering Research Council of Canada, the Kidney Foundation of Canada, and the Vanier Canada Graduate Scholarship for their funding support. The authors thank Tim Salcudean for infrastructure and support.

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
