# OpenReview forum: "Speckle and Shadows: Ultrasound-specific Physics-based Data Augmentation Applied to Kidney Segmentation"
_MIDL.io/2022/Conference — MIDL 2022_

### Official Review · Reviewer_VkBT · 2022-01-07

**Confidence:** 4
**Preliminary Rating:** 2

**Summary:**

“Speckle and Shadows” presents three image augmentation techniques making use of ultrasound physics, with a particular emphasis on improving segmentation performance in the full and semi-supervised settings. Experiments are performed on an in-house dataset on a multi-structure kidney segmentation task. The proposed techniques are not shown to consistently improve performance in either setting, as detailed below.

**Strengths:**

- The paper is well motivated as modality-specific inductive prior-driven augmentations have been shown to improve performance in other imaging sources.
- The paper is well organized and written as I was immediately able to understand the methods of the work based on a preliminary skim prior to detailed reading.
- The explanations of how the parameters of the proposed augmentations are tuned are extensive.
- Experiments are presented on both fully supervised segmentation and semi-supervised representation learning-driven segmentation.

**Weaknesses:**

Disclosure: My criticisms will focus on the scientific aspects of this submission and not its place in the literature/usefulness to the ultrasound research community as I am not familiar with the ultrasound segmentation literature.

While the paper has an interesting motivation, unfortunately, I do not believe that the paper is currently ready for publication, based on the following:

- **Unclear contribution**. One of the three proposed augmentations is “zoom”ing, which appears to be plain image resizing from the description. Random image resizing is universally used in vision applications (and even ultrasound segmentation applications from 2016 [Baumgartner17]) – please clarify the contribution of “zoom” augmentation in this work and please correct me if I'm wrong.

- **Missing baseline comparisons**. The paper cites [Lee21] and [Tirindelli21] as works also developing ultrasound-specific augmentations but only compares against versions of itself, which is more of an ablation study. Please present experimental results against other works in this subfield so as to properly contextualize the contribution and utility of these augmentations or explain why they are inapplicable to your data/task.

- Please clarify if I misunderstood, but to find the parameters for the proposed “acoustic shadow” augmentation, ground truth shadow segmentation masks are required. This leads to a **subtle train set leakage** into the semi-supervised settings: the final contrastive network may be finetuned with only 1% of the data, but the augmentations used to pretrain the network needed a much higher percentage to find their optimal parameters.

- **Performance degradation using the proposed techniques**.
  * Segmentation performance actually reduces in Table 1 (nnUNet experiments) with the incorporation of the proposed techniques. In Table 2, only the 100% data availability setting of [Chaitanya20]+proposed techniques makes a small improvement over baseline with the other data availability settings in Table 2 showing degradation with use. Table 1 performance is explained by “nnUnet adapting specifically to the dataset to optimize performance”, which is confusing. If the proposed augmentations are useful for this dataset, wouldn’t nnUnet adapt to their usage for optimal results? Please clarify.
  * The proposed shadowing augmentation consistently degrades segmentation performance and the paper states that it can be optimized. I do not think that “shadowing” is ready for publication until these issues are resolved as neither positive nor negative conclusions can be drawn about its use.

- **Unclear configurations of baseline networks**. [Chaitanya20] is a method developed for affine-aligned volumetric images, whereas, from what I understand, it is trained on 2D+time videos in this submission. Please explain how the original framework is extended to this use case. Moreover, both baselines [Chaitanya20, Isensee21] make use of extensive augmentation routines with commonly used transformations. Were these enabled or turned off for the experiments in the paper?

- **Unclear experimental design**. Only 2 out of 4 proposed augmentation configurations are used when evaluating nnUnet + proposed transformations (missing “speckle” and “all”) – why?

- The paper develops two forms of speckle augmentation (scale and shape), but then **drops one of them** based on a mutual information-based heuristic. While I’ve seen MI heuristics be used for active learning and hyperparameter optimization in medical imaging [Nath21], I have not seen them used for choosing augmentations and the provided explanation (drop the one that has higher MI to the original because of redundancy) is not immediately convincing, as several standard high-performing augmentations yield images with very high MI to their inputs. Please expand on this.

- In several instances, the paper claims that various “classic” transformations which produce unrealistic augmented images hurt segmentation performance. This **claim needs to be experimentally validated** as unrealistic transforms (cutmix, cutout, mixup, flips, etc.) often do improve segmentation/representation learning performance counter-intuitively in both natural [French20] and medical images [Chen19, Chaitanya20]. In fact, nnUnet [Isensee21], which is used here as an upper bound on performance, by default uses mirroring along all directions which is explicitly stated to be suboptimal in this submission. From another perspective, even if a dataset may not be globally symmetric w.r.t. some transform (e.g. 90-degree rotation as mentioned in the submission), local symmetries are often still useful towards downstream applications like segmentation or detection [Winkels19]. Lastly, one of the proposed augmentations (“speckle”) creates a parameter map and not a realistic ultrasound image, thereby contradicting the paper's motivation. I believe that the parts of this paper relevant to augmentation realism need to be better motivated or removed.


References
---------------
[Baumgartner17] Baumgartner, Christian F., et al. "SonoNet: real-time detection and localisation of fetal standard scan planes in freehand ultrasound." IEEE transactions on medical imaging 2017.

[Chen19] Chen, Liang, et al. "Self-supervised learning for medical image analysis using image context restoration." Medical image analysis 2019.

[Winkels19] Winkels, Marysia, and Taco S. Cohen. "Pulmonary nodule detection in CT scans with equivariant CNNs." Medical image analysis 2019.

[Chaitanya20] Chaitanya, Krishna, et al. "Contrastive learning of global and local features for medical image segmentation with limited annotations." NeurIPS 2020.

[French20] French, Geoff, et al. "Semi-supervised semantic segmentation needs strong, varied perturbations.” BMVC 2020.

[Isensee21] Isensee, Fabian, et al. "nnU-Net: a self-configuring method for deep learning-based biomedical image segmentation." Nature methods 2021.

[Lee21] Lee, Lok Hin, Yuan Gao, and J. Alison Noble. "Principled Ultrasound Data Augmentation for Classification of Standard Planes." IPMI 2021.

[Tirindelli21] Tirindelli, et al. “Rethinking Ultrasound Augmentation: A Physics-Inspired Approach”, MICCAI 2021.

[Nath21] Nath, Vishwesh, et al. "The Power of Proxy Data and Proxy Networks for Hyper-Parameter Optimization in Medical Image Segmentation." MICCAI 2021.


**Deanonymize Review:**

no

**Detailed Comments:**

- The narrative explanation of the acoustic shadowing augmentation algorithm is quite unclear. Please create a visual representation of this process to aid in understanding (e.g. a block diagram with accompanying figures).

- Typos here and there: (1) Sec 2.2, par. 1, after the reference to Hu 2019a; (2) the last sentence of page 5.


**Final Rating After The Rebuttal:**

2: Weak Reject

**Justification Of The Final Rating:**

During the rebuttal, the submission was very admirably revised from its initial state and I thank the authors for their responses. I summarize the paper and rebuttal as:

1. The paper proposed a few ultrasound-specific image augmentations, 3 in the original submission and another new one in the rebuttal.
2. The paper's presentation and claims were properly adjusted based on reviewer feedback.
3. In its experiments on a single dataset, the augmentations were found to have rather inconsistent trends (sometimes helping, sometimes hurting). A good illustration of this is in Table 2 (col 7), which shows no discernible pattern between the data availability, augmentation, and downstream segmentation performance.
4. To address this aspect, the rebuttal added statistical significance analysis to the comparisons. However, significant performance improvements/degradations are isolated to individual pairwise comparisons and there is still no overall trend.
5. Due to rebuttal time constraints, comparisons against relevant ultrasound augmentation papers could not be performed.
6. Finally, the experiments using "all" augmentations were removed and the stated reason of [Lee21] finding a better combination of augmentations did not make immediate sense to me. In practice, if these augmentations are useful, practitioners would use all of them anyway, even if they are not precisely optimized.

As a result, I do not think that the paper is yet ready for publication as no conclusion (positive or negative, would be happy to see either) can be drawn from its current findings. I do not know if the lack of a trend is because this dataset is peculiar in some manner or whether other existing works would reveal a stronger trend. I highly encourage that future revisions of this work to compare against relevant literature in its subdomain and include results on more datasets.

(Note to the AC: if there was a 'borderline reject' rating option, I would use that instead of 'weak reject')

**Paper Type:**

methodological development

**Questions To Address In The Rebuttal:**

Please see the weaknesses above. In order to clarify the utility of the proposed methods, I believe the following experiments/clarifications are required:
- Is "zoom" augmentation just image resizing?
- Relevant work on ultrasound data augmentation needs to be benchmarked against.
- The experiments with the subtle train set leakage into the semi-supervised settings need to be rectified. Perhaps discarding the images used to estimate Nakagami parameters from all subsequent experiments would work?
- If performance consistently degrades with the use of the proposed techniques in a self-tuning supervised segmentation framework and a contrastive semi-supervised framework (Table 1 & 2 experiments), please further clarify as to why the proposed augmentations should be used.
- Please further clarify the experimental configurations mentioned in the weaknesses.
- Please include the other form of speckle augmentation in our experiments as well or better rationalize its exclusion.
- Please demonstrate that unrealistic augmentations do actually harm segmentation performance or remove these statements.
- This is probably too much for a rebuttal, but inconsistent experimental trends can often be clarified by experiments on more than one dataset and I believe that there are several public and open ultrasound segmentation datasets.

**Special Issue:**

no

---

### Official Review · Reviewer_ahuw · 2022-01-23

**Confidence:** 4
**Preliminary Rating:** 4
**Recommendation:** Poster

**Summary:**

The paper presents three types of image data augmentation introduced to specifically enhance the perfomance of ultrasound image segmentation. A zoom feature is introduced (keeping the beam cone intact), a speckle feature, and a shadow feature. The approach is evaluated on a kidney data sets, using two baseline methods for comparison, the nnUnet and the contrastive learning approach from Chaitanya. For some tissue classes the proposed approach shows moderate improvements. The paper lacks detail in the presented approaches.

**Strengths:**

The authors compare their appraoch against to state of the art approaches, one, the nnUnet as a fully supervised approache, the other, the contrastive learning approach from Chaitanya as a semi-supervised approach.
I find the shadow based augmentation especially interesting since it is a typical ultrasound feature producing substantial difficultieds for any segmentation algorithm. It seems that the approach is not completely new, Xu et al. used shadow based augmentation for prostate segmentation (X. Xu, T. Sanford, B. Turkbey, S. Xu, B. J. Wood and P. Yan, "Shadow-consistent Semi-supervised Learning for Prostate Ultrasound Segmentation," in IEEE Transactions on Medical Imaging, doi: 10.1109/TMI.2021.3139999). It is very instructive to see this idea applied, even if the experimental results are a bit disapointing here.

**Weaknesses:**

Idea and approach of the paper is good, however, it is a bit disapointing that the in comparision with the nnUnet baseline, only the zoom feature produced an advantage and a relatively moderate one. In comparision to the Chaitanya approach, all the new introduced features led to some improvements, however, pretty scattered over the examined tissue types and percentage of label data used. The authors could not show that the extended augmentation is of consistend advantage. It is especially disturbing, that the extended augementation led in perhaps half of the cases to a deterioration of the segmentation results.
The paper is a bit vaguely written, for the reader it would be very difficult to reproduce the experiments.

**Deanonymize Review:**

no

**Detailed Comments:**

The implemented methods are not described in sufficient detail.
- Was the nnUnet applied to the image matrix as is, or to a resampled matrix (automatically determined following the standard aproach of the nnUnet)?
- How was the image cone retained in the zoom feature? Zooming and setting the image values outside the cone to zero?
- In ultrasound images, shadows are produced due to a sudden change in impedance of the tissue. Was this used to position the artificial shadows at somewhat realistic locations? Was the shadow region just replaced by sampling the Nakabami distribution or was it mixed with the original image content? The procedure were and how exactly the shadows was placed should be described in more detail, a formula of how the pixel intensity was set would be helpful. Please add also example images of the simulated shadow regions.

**Paper Type:**

methodological development

**Questions To Address In The Rebuttal:**

- why does the extended augmentation lead in many of the examined settings to deterioration? Did the authors perform any experiments towards identifying possible reasons?
- How was the augmentation exactly introduced? Did the training set grow with the (additional) augmentation, or did they replace the previously used set?
- Desribe in detail how the the artifical shadows were introduced (please supply a formula for the replaced pixels).

**Special Issue:**

no

---

### Official Review · Reviewer_GKtE · 2022-01-24

**Confidence:** 4
**Preliminary Rating:** 3
**Recommendation:** Poster

**Summary:**

The paper examines physically based data augmentation schemes for kidney segmentation in ultrasound. The augmentations include variations on zooming, acoustic shadowing,  and speckle parameter maps. The augmentation schemes are used in segmentation of four classes, kidney capsule, cortex, medullary and central echogenic complic and it is validated on 514 images. They claim benefits of augmentation are demonstrated.


**Strengths:**

- Addresses the important problem of augmentation and which augmentations to use. In medical imaging, augmentations are often approached with a set of highly unrealistic transformations that are generic for all problems. Moreover, augmentations, while common place, often fail to provide tangible benefits, so it is good to see more work examining domain specific augmentations.

- Clearly defined experiment, a small number of sensible realistic augmentations are tested on relevant data to investigate their relevance using otherwise established algorithms.

- The work is well written, language is appropriate, and adequately addresses prior work.

**Weaknesses:**

- The results are too a large degree negative in the sense that the proposed augmentations do not appear to provide any benefits. They also do not appear to lead to worse results overall. Statistical tests are not done, but I find it hard to see a consistent pattern. Negative results can and should be published in many cases and in this case it actual could be, as the experiment and approach makes sense.

- Too positive reporting of results. I find the description of the results to be too positive. For instance they claim in the Conclusion that the results are promising, in the discussion they claim the augmentations can help (first sentence), and in the abstract that result demonstrate the benefit of them. Why not use the opportunity to expand on the difficulties of doing augmentations right?


**Deanonymize Review:**

no

**Detailed Comments:**

- I fail to understand how imaging artifacts become positive contributors, as mentioned as point (iii) in the beginning of methods. Augmentation with artifacts helping to make methods invariant to artifacts is not the same as real imaging artifacts becoming a positive contributor. I don't think the authors are claiming artifacts will help performance?

- I am not familiar enough with ultrasound to speak to the correctness of the method used to generate artificial ultrasound shadows and the speckle parameter maps.

- "As this network optimizes on the data set, it represents a reasonable upper-bound on performance.", I find it unclear what is meant by this. I assume independent test sets were used to estimate performance.


**Final Rating After The Rebuttal:**

4: Weak Accept

**Justification Of The Final Rating:**

I thank the authors for the revision. I happy that the reporting is now more accurate and negative results are described as such. I think the experiment makes sense to publish even with negative results, hence and update of the score to weak accept.

**Paper Type:**

methodological development

**Questions To Address In The Rebuttal:**

- I would like the authors to address the questions about the significance and usefulness of their augmentations. Is it really correct to say that your results demonstrate benefit of the speckle maps and zoom augmentations? How do you determine this?

**Special Issue:**

no

---

### Meta-Review · Area_Chair_FYLU · 2022-02-13

**Recommendation:** Accept (Poster)
**Confidence:** 4

**Metareview:**

I would like to thank all reviewers for their time and effort spent reviewing this paper and for engaging with the authors’ detailed rebuttal. On balance, I think that there is sufficient consensus to accept the paper. But I would strongly encourage the authors to try to address, as best they can, the remaining comments in the camera-ready version of the paper. Specifically, if time permits, I would like to see the authors try to include some improved comparative evaluation as suggested by Reviewer VkBT.

---

### Decision · Program_Chairs · 2022-02-28

Accept